

# Mathematical model explains differences in Omicron and Delta SARS-CoV-2 dynamics in Caco-2 and Calu-3 cells

Vladimir Staroverov[1], Alexei Galatenko[1,2], Evgeny Knyazev[2,3] and Alexander Tonevitsky[2,3,4]

[1] Faculty of Mechanics and Mathematics, Lomonosov Moscow State University, Moscow, Russia
[2] Faculty of Biology and Biotechnology, HSE University, Moscow, Russia
[3] Shemyakin-Ovchinnikov Institute of Bioorganic Chemistry, Russian Academy of Sciences, Moscow, Russia
[4] Art Photonics GmbH, Berlin, Germany

Corresponding author
Evgeny Knyazev, eknyazev@hse.ru

## ABSTRACT

Within-host infection dynamics of Omicron dramatically differs from previous variants of SARS-CoV-2. However, little is still known about which parameters of virus-cell interplay contribute to the observed attenuated replication and pathogenicity of Omicron. Mathematical models, often expressed as systems of differential equations, are frequently employed to study the infection dynamics of various viruses. Adopting such models for results of *in vitro* experiments can be beneficial in a number of aspects, such as model simplification (*e.g.*, the absence of adaptive immune response and innate immunity cells), better measurement accuracy, and the possibility to measure additional data types in comparison with *in vivo* case. In this study, we consider a refinement of our previously developed and validated model based on a system of integro-differential equations. We fit the model to the experimental data of Omicron and Delta infections in Caco-2 (human intestinal epithelium model) and Calu-3 (lung epithelium model) cell lines. The data include known information on initial conditions, infectious virus titers, and intracellular viral RNA measurements at several time points post-infection.
The model accurately explains the experimental data for both variants in both cell lines using only three variant- and cell-line-specific parameters. Namely, the cell entry rate is significantly lower for Omicron, and Omicron triggers a stronger cytokine production rate (*i.e.*, innate immune response) in infected cells, ultimately making uninfected cells resistant to the virus. Notably, differences in only a single parameter (*e.g.*, cell entry rate) are insufficient to obtain a reliable model fit for the experimental data.

# INTRODUCTION

Coronavirus disease 2019 (COVID-19), a disease caused by the severe acute respiratory syndrome coronavirus 2 (SARS-CoV-2), is a systemic infection affecting various human tissues and organs (*El-Kassas et al., 2023*). The primary route for virus entry into human cells is through the interaction between the spike (S) protein and angiotensin-converting

enzyme 2 (ACE2) (*Kirtipal et al., 2022*). The main clinical symptoms of COVID-19 are typically attributed to lung cell damage. However, RNA sequencing analysis of different human organ samples has identified high ACE2 expression levels in the gastrointestinal tract, alongside serine protease TMPRSS2, which is also essential for SARS-CoV-2 cellular entry (*Nersisyan et al., 2020*). This explains digestive tract involvement and symptoms *via* direct viral impact, immune response, and inflammation (*Chen et al., 2022*; *Chu et al., 2021*). Individuals infected with the Omicron variant exhibited increased amounts of SARS-CoV-2 RNA in anal swabs, while gastrointestinal symptoms were less frequent compared to the Delta variant (*Menni et al., 2022*; *Shi, Mei & Wang, 2022*). These observations warrant exploring Omicron's dynamics not only in the lung cells but also in the intestinal epithelium, aided by the Caco-2 cell line expressing ACE2 and TMPRSS2 (*Knyazev, Nersisyan & Tonevitsky, 2021*).

The general importance of mathematical modeling of biological and pathological processes cannot be overstated. Various scientific disciplines generate copious quantitative data that necessitate processing and analysis through mathematical models to formulate and test hypotheses effectively (*Vera et al., 2021*). Historically, mathematical modeling has been pivotal across evolutionary biology, structural biology, and biochemistry, especially in understanding enzymatic reactions and allosteric regulation. Moreover, mathematical modeling aids in evaluating the pharmacokinetics and pharmacodynamics of potential drugs and helps decipher the interrelationships of genes, nucleic acids, proteins, lipids, and other molecules (*Vera et al., 2021*). Mathematical models serve the purpose of either encompassing multiple factors to replicate studied processes with precision or simplifying and isolating individual components to elucidate their specific roles in the broader context (*Edelstein-Keshet, 2005*).

In virology, mathematical modeling is utilized to investigate socially significant viruses like hepatitis, HIV, Ebola, Zika, and coronaviruses. Employing various equations—such as ordinary, delayed, partial, and integro-differential equations—enables the exploration of viral infection dynamics within subcellular compartments, cells, organisms, and entire populations. This approach illuminates the roles of diverse molecules, complexes, signaling pathways, and immune components (*Hattaf & Yousfi, 2020*). The virus intracellular cycle encompasses intricate biochemical reactions and transport processes. When represented in mathematical models, it provides insights into potential antiviral drug targets and the dynamics of virus life cycles (*Grebennikov et al., 2021*). Mathematical modeling, spanning various scientific domains, consistently generates valuable insights and new hypotheses, prompting researchers to challenge established paradigms (*Layden et al., 2003*).

Epidemiologists worldwide have conducted extensive studies on SARS-CoV-2 transmission dynamics and mitigation strategies, with mathematical modeling proving crucial in quantifying influential determinants (*Ferretti et al., 2020*; *Endo et al., 2020*; *Dickens et al., 2020*; *Li et al., 2020*), aiding in designing effective control measures. Studies on within-host infection dynamics using mathematical models enable the optimization of

randomized clinical trial designs for antiviral treatments, as demonstrated by the viral dynamics model (*Iwanami et al., 2021*). Additionally, these studies contribute to the revision of guidelines for COVID-19 patient isolation (*Jeong et al., 2021*) and the estimation of the COVID-19 incubation period, utilizing viral load data to mitigate the risk of disease spread (*Ejima et al., 2021*). Meanwhile, mathematical models applied to experimental SARS-CoV-2 infections in different cell lines suggest disparities in viral replication cycles, indicating limitations in drug screening assays beyond a day post-infection in some cell lines, necessitating further quantitative investigations for therapeutic development (*Bernhauerová et al., 2021*). This knowledge is pivotal in early antiviral treatment development, aiding in the assessment of therapeutic effectiveness and optimizing dosage and administration timings (*Bernhauerová et al., 2021*). Mathematical models for SARS-CoV-2, while not necessarily overly complex, serve as vital representations of the virus-human interaction. Moving from descriptive to predictive models necessitates collaboration between modelers and clinical experts to address COVID-19's complex dynamics, aiding in developing effective tools for understanding and managing its pathogenesis and treatment (*Grebennikov et al., 2022*).

The SARS-CoV-2 infection models can be classified based on different interactions between the virus and the host organism, encompassing virus spreading in cells, tissues, and organs, interactions with the innate immune response, engagement with the adaptive immune system, combined effects on both innate and adaptive immunity, and the comprehensive immunophysiological responses of the host, involving various inflammatory and physiological mechanisms (*Grebennikov et al., 2022*). Our model, developed from experimental data on two SARS-CoV-2 variants within monocultures of two cell types, specifically excludes the acquired immune response and innate immunity cells from consideration. This exclusion enables us to assess the significance of individual virus life cycle events and innate immunity components within cell models representing the extensive human epithelial tissues of the intestine and lung, known for harboring SARS-CoV-2 receptors. The developed model holds the potential to predict alterations in SARS-CoV-2 behavior within cells when specific viral properties change. For instance, recent observations of a new variant, BA.2.86 (pirola), emerging in late summer 2023, demonstrated a resurgence in its ability to effectively infect lung cells by utilizing TMPRSS2 for cell entry (*Zhang et al., 2024*). This resurgence parallels earlier variants like Alpha, Beta, Gamma, and Delta, urging prompt investigation through mathematical models and experimental data.

We previously introduced a mathematical model elucidating SARS-CoV-2 infection dynamics in Caco-2 cells using *in vitro* data. Our study compared Delta variant dynamics to the wild-type virus and revealed significant differences in viral cell entry rate and cytokine production rate between two variants (*Staroverov et al., 2023*). The Omicron variant of SARS-CoV-2 appeared in November 2021, quickly replaced all other variants, and drastically changed the pathogenesis of COVID-19 (*Chatterjee et al., 2023*). The

Omicron variant shows reduced utilization of the TMPRSS2 protease compared to earlier variants, potentially explaining its lowered cellular entry efficiency and concurrent clinical manifestations (*Meng et al., 2022*; *Hu et al., 2022*; *Chan et al., 2022*; *Shuai et al., 2023*). Furthermore, variations in cytokine secretion profiles among different SARS-CoV-2 variants may further contribute to the complexity of their clinical phenotypes (*Singh et al., 2023*).

The precise contribution of reduced viral entry *vs* increased innate immune response in shaping the infectious phenotype of the Omicron variant remains unclear. In this study, we applied our previously developed mathematical model (*Staroverov et al., 2023*) to delineate the relative importance and interplay between these two factors. The model was fit to the *in vitro* data collected from infected Caco-2 cells, an established *in vitro* model of the human intestinal epithelium, and Calu-3 cells used to model SARS-CoV-2 infection in lung epithelium.

## MATERIALS AND METHODS

### Experimental data

The primary experimental dataset was derived from the recent study for the Caco-2 and Calu-3 cells (*Shuai et al., 2022*). At t = 0 point, the authors added various SARS-CoV-2 variants to 30,000 Caco-2 cells at a multiplicity of infection (MOI) of 0.1 and to 30,000 Calu-3 cells at an MOI of 0.5, followed by a 2-h incubation period. Subsequently, they washed the cells to remove non-entered virions and replaced the culture medium. SARS-CoV-2 infectious titers were then determined using $TCID_{50}$ assays at three time points: t = 8, 24, and 48 h post-infection (hpi). *GAPDH*-normalized expression of subgenomic viral envelope RNA (*sgE*) was determined using RT-qPCR at four time points (t = 2, 8, 24, and 48 hpi). The RT-qPCR data encompassed intracellular viral RNA abundance without genomic RNA present within virus particles.

The modeling of virus dynamics in cells considered the phenomenon that both Caco-2 and Calu-3 cells remained viable and showed no cytopathic effects for at least 48 h after infection with both Omicron and Delta variants (*Mautner et al., 2022*; *Dighe et al., 2022*). Furthermore, the proportion of uninfected healthy cells during the experiment remained relatively high (*i.e.*, the virus did not affect the entire cell culture), as confirmed by several studies on the infection dynamics of Caco-2 and Calu-3 cells with various SARS-CoV-2 variants (*Chu et al., 2020*; *Shuai et al., 2020*; *Bojkova et al., 2022b*; *Bahlmann et al., 2023*).

### Mathematical model

Our model is in essence based on our previously developed continuous model (*Staroverov et al., 2023*) which was inspired by a discrete model. It is comprised of healthy ($U_0$), infected ($I$), and resistant ($U_1$) cells, cytokines ($Cyt$), free virions ($V$), and intracellular viral RNA ($R$). Due to the discrete origin all these constituents are measured in pieces. We consider the following system of integro-differential equations:

$$\begin{cases} U_0(t)' = -\beta U_0(t)V(t) - \beta_{cyt}Cyt(t)U_0(t) & (1) \\[8pt] U_1(t)' = -\frac{\beta}{k}U_1(t)V(t) + \beta_{cyt}Cyt(t)U_0(t) & (2) \\[8pt] Cyt(t)' = p_c I(t - \Delta_{tcyt}) - \beta_{cyt}Cyt(t)U_0(t) & (3) \\[8pt] I(t)' = \beta(U_0(t) + U_1(t)/k)V(t) & (4) \\[8pt] V(t)' = \int\limits_0^t (\beta(U_0(t-x) + U_1(t-x)/k)V(t-x))p(x)dx - \\[6pt] \qquad\quad -\frac{\beta}{k}(I(t) + U_1(t))V(t) - \beta U_0(t)V(t) & (5) \\[8pt] R(t) = \int\limits_0^t (\beta(U_0(t-x) + U_1(t-x)/k)V(t-x))\sigma(x)dx & (6) \end{cases}$$

As evident from the equation system (Eq. (1)), the infection intensity for healthy cells is directly proportional to both the number of healthy cells and the count of free virions; healthy cells undergo a transition to a resistant state with an intensity that correlates with both the number of healthy cells and the quantity of cytokines (Eq. (1)). Resistant cells are generated from healthy cells (the first summand of Eq. (2)) and get infected with intensity proportional to the product of the number of resistant cells and the number of free virions (note that the intensity parameter $\beta$ from the first equation here is divided by the parameter $k$, *i.e.*, intensity is reduced). Cytokines are generated by infected cells with some delay and are used to make healthy cells resistant (Eq. (3)). Infected cells are produced either from healthy cells or from resistant cells (Eq. (4)). Free virions are generated by infected cells with a certain density $p$ (the first summand of Eq. (5)) and can enter both healthy and already infected cells (the second summand of Eq. (5)). Finally, intracellular viral RNA is produced by infected cells with a density $\sigma$.

Note that within the model framework, the units are consistent. For example, when a virion infects a healthy cell, both $U_0$ and $V$ are simultaneously reduced by a uniform value of 1. Additionally, intracellular viral RNA ($R$) and Eq. (6) were added to the model solely for the purpose of parameter fitting. The function $R(t)$ is not included in the first five equations of the system.

### Computation of density functions

Of particular interest are the densities $p$ and $\sigma$. The function $p(t)$ corresponds to the intensity of virus generation by infected cells; the function $\sigma(t)$ is the intensity of intracellular RNA generation. Our model of virus generation intensity takes into account the delay required to start virus production and internal cellular resources used to produce viral particles (*i.e.*, when the resources are exhausted, production is stopped). Justification of this model was presented in *Staroverov et al. (2023)*. However, the detailed and explicit model of the function $p(t)$ used in *Staroverov et al. (2023)* turned out to be computationally hard. To speed up computations, we chose to replace the detailed model with a delayed $\delta$-function, a method also employed in *Ghosh, Volpert & Banerjee (2023)*, *i.e.*, to assume that all resources are used "at once" and all viruses produced by an infected cell are

generated instantly at the time moment $t_0 + \Delta_{t\,latent}$, where $t_0$ is the time of infection. As it was noticed in *Ghosh, Volpert & Banerjee (2023)* and our previously conducted parameters sensitivity analysis (*Staroverov et al., 2023*), such an assumption does not lead to essential model precision degradation. As a result, Eq. (5) of the system above took the form

$$V(t)' = (\beta(U_0(t - \Delta_{t\,latent}) + U_1(t - \Delta_{t\,latent})/k) \cdot V(t - \Delta_{t\,latent})) \cdot L_0/n_{l2V} -$$
$$- \frac{\beta}{k}(I(t) + U_1(t))V(t) - \beta U_0(t)V(t),$$

*i.e.*, the integral on the right-hand side was removed. The detailed description of model parameters used throughout this section is presented in the subsequent section.

On the contrary, the function $\sigma$ was modeled in more detail: we considered viral RNA used to produce viral particles. Similarly to *Staroverov et al. (2023)*, it is computed at time points with a constant step $\Delta_h$ using the following formula. Initially, *i.e.*, at the moment of infection ($t = 0$), $\sigma = 1$. Next, at time point $i \cdot \Delta_h$

$$\sigma(i) = \begin{cases} \sigma(i-1) \cdot 2^{\Delta_h/\Delta_{tRNAdouble}}, & \text{if } \Delta_{t\,latent} \notin (\Delta_h \cdot (i-1), \Delta_h \cdot i], \\ \max\left(0, \sigma(i-1) \cdot 2^{\Delta_h/\Delta_{tRNAdouble}} - n_{RNA2V} \cdot L_0/n_{l2V}\right) & \text{otherwise.} \end{cases}$$

In other words, intracellular viral RNA grows exponentially with the rate specified by the parameter $\Delta_{t\,RNAdouble}$. Additionally when the virus emission occurs, RNA is used to produce virions. Since the emission in our model is instant, the quantity of RNA required to produce all viruses is subtracted at the corresponding mash point.

The values at the remaining points are recovered using linear interpolation.

Note that $\sigma$ only needs to be computed at four points of internal viral RNA measurement and its evaluation at each point requires constant complexity.

### Model parameters

The parameters of the model are listed in Table 1. The parameter inference procedure, which was identical to the one used in *Staroverov et al. (2023)*, is outlined below.

The model uses four types of parameters. Fixed parameters are taken from the literature or directly extracted from the data. Global parameters have equal values for all cell lines and virus variants. Local parameters may depend on both cell line and virus variant identity. Global and local parameters are inferred from the data. In particular, the parameters $Norm_V$ and $Norm_{RNA}$ specify the scaling coefficients for transforming the experimental data (the results of $TCID_{50}$ assays and *GAPDH*-normalized expression of subgenomic viral envelope RNA, respectively) into the model units, *i.e.*, pieces. Note that $Norm_V$ is strain-dependent (but independent of the cell line), whereas $Norm_{RNA}$ is the same for all variants. Internal parameters are related to the implementation of the model. $\Delta_{DE}$ specifies the time step in the numerical solution procedure. The parameters $\alpha_i$ and $\alpha'_j$ are the weights in the error functional (see the subsequent Section for detail).

In *Staroverov et al. (2023)*, it was shown that the pair of parameters $(L_0, n_{l2V})$ is dependent, meaning that modification of one parameter can be compensated by modifying the other. Thus, the parameter $L_0$ was set equal to 1. The remaining parameters were proved to be independent.

**Table 1 Parameters of the models (pcs stands for pieces).**

| Parameter | Type | Meaning (units, if applicable) | Value (confidence interval, if applicable) |
|---|---|---|---|
| $\Delta_{t\,clean}$ | Fixed | Time to washing (h) | 2 |
| $L_0$ | Fixed | Initial resource concentration (unit) | 1 |
| $\Delta_{t\,cyt}$ | Fixed | Time from cell infection to cytokine generation (h) | 5 |
| $C_{initial}$ | Fixed | Initial number of healthy cells (pcs) | 30,000 |
| $V_{initial}$ | Fixed | Initial number of virions (pcs) | Caco-2 3,000; Calu-3 15,000 |
| $N_V$ | Fixed | The number of points in time for free virion measurements (pcs) | 3 |
| $N_{RNA}$ | Fixed | The number of points in time for viral RNA measurements (pcs) | 4 |
| $\tau_1$ | Fixed | Time corresponding to the first measurement of infected cells (h) | 2 |
| $\Delta_{t\,RNA\,double}$ | Fixed | Time required to double viral RNA in an infected cell (h) | 20 |
| $\beta_{cyt}$ | Global | Intensity of cytokine-cell interactions $((pcs \cdot h)^{-1})$ | $1.18025 \cdot 10^{-8}\ [8.1 \cdot 10^{-9}, 1.6 \cdot 10^{-8}]$ |
| $n_{l2V}$ | Global | Resources used to produce a single virion (fraction of the relative unit) | $0.000067\ [0.00006, 0.00008]$ |
| $n_{RNA2V}$ | Global | RNA used to produce one viral particle (pcs) | $6.5 \cdot 10^{-5}\ [6.0 \cdot 10^{-5}, 7.3 \cdot 10^{-5}]$ |
| $\Delta_{t\,latent}$ | Global | Time from cell infection to virus generation (h) | $7.36095\ [6.4, 7.7]$ |
| $Norm_{RNA}$ | Global | Normalization coefficient for the concentration of viral RNA | $0.74\ [0.41, 0.77]$ |
| $Norm_V$ | Local | Normalization coefficient for the number of virions | Delta: 1.57 [1.54, 1.90]; Omicron: 2.1 [1.95, 2.26] |
| $\beta$ | Local | Intensity of cell infection $((pcs \cdot h)^{-1})$ | Caco-2, Delta: $1.0203 \cdot 10^{-8}\ [9.4 \cdot 10^{-9}, 1.1 \cdot 10^{-8}]$; Caco-2, Omicron: $7.73966 \cdot 10^{-9}\ [6.6 \cdot 10^{-9}, 9.1 \cdot 10^{-9}]$; Calu-3, Delta: $1.95238 \cdot 10^{-8}\ [1.8 \cdot 10^{-8}, 2.3 \cdot 10^{-8}]$; Calu-3, Omicron: $4.56125 \cdot 10^{-9}\ [3.8 \cdot 10^{-9}, 5.4 \cdot 10^{-9}]$ |
| $p_c$ | Local | Cytokine generation intensity $((pcs \cdot h)^{-1})$ | Caco-2, Delta: 63,108.2 [43,346., 88238.]; Caco-2, Omicron: 482,281 [395,055., 582,922.]; Calu-3, Delta: 65,066.7 [52,742., 78,995.]; Calu-3, Omicron: 350,478 [287,090., 425,733.] |
| $k$ | Local | Reduction of infection intensity for infected and resistant cells (times) | Caco-2, Delta: 37.9835 [28, 52]; Caco-2, Omicron: $10^9$ or $+\infty$ [$1.7 \cdot 10^8, +\infty$]; Calu-3, Delta: 138.09 [114, 244]; Calu-3, Omicron: $10^9$ or $+\infty$ [$1.8 \cdot 10^8, +\infty$] |
| $\Delta_{DE}$ | Internal | Step in numerical solution of the system of equations (h) | $10^{-3}$ |
| $\Delta_h$ | Internal | Step in evaluation of the function $p$ (h) | $10^{-5}$ |
| $\alpha_i, i = 1, \ldots, N_V$ | Internal | Weights for penalties in the error functional | $1, 10, 10$ |
| $\alpha'_j, j = 1, \ldots, N_{RNA}$ | Internal | Weights for penalties in the error functional | $1, 5, 40, 60$ |

## Selection of parameter values

The values of fixed and internal parameters were set to the values reported in *Staroverov et al. (2023)*. Similarly to *Staroverov et al. (2023)*, global and local parameters were selected using the coordinate descent method with respect to the following error functional depending on the model parameters:

$$Err(Params) = \sum_{i=1}^{N_V} \alpha_i (V_i - Norm_V \cdot v_i)^2 + \sum_{j=1}^{N_{RNA}} \alpha'_j (R_j - Norm_{RNA} \cdot r_j)^2$$

Here $v_i$ is the average of $\log_{10}$ of the number of virions in the experiment at the $i$th measurement, $V_i$ is $\log_{10}$ of the number of free virions in the model at the corresponding point of time, $r_j$ is the average of $\log_{10}$ of the concentration of viral RNA in the experiments at the $j$th measurement, $R_j$ is $\log_{10}$ of the value of viral RNA concentration in the model at the corresponding point of time. The weights $\alpha_i$ and $\alpha'_j$ were selected to compensate the variance of the experimental data and to reduce the influence of early time points, *i.e.*, we assign smaller weights to small numbers.

Confidence intervals for the inferred parameters were evaluated in the following way. Similarly to *Staroverov et al. (2023)*, we changed the value of the parameter of interest (*i.e.*, decreased it to obtain the left interval boundary and increased it to obtain the right one) until the value of the error functional *Err* grew by 5%.

## RESULTS

### Overview of the experimental data and the mathematical model

We begin with the overview of the experimental longitudinal data previously generated and described in *Shuai et al. (2022)*. In Fig. 1, we present the experimental measurements for 2 × 2 design, where we studied the Delta and Omicron variants in two distinct cell lines: Caco-2 (Figs. 1A and 1B) and Calu-3 (Figs. 1C and 1D). Each combination of virus variant and cell line includes measurements of infectious virus titers at three time points (left panel) and the abundance of subgenomic intracellular SARS-CoV-2 RNA at four time points (right panel). Furthermore, it is important to note that prior studies have consistently shown that nearly 100% of Caco-2 and Calu-3 cells remained viable without exhibiting visible cytopathic effect for at least 48 h of the experiment, with a fairly high fraction of cells staying uninfected (for more details, refer to the Materials and Methods section).

In our previous work (*Staroverov et al., 2023*), we developed a mathematical model to describe the infection dynamics of both the wild-type and the Delta variants in Caco-2 cells. The primary experimental data used in this study was obtained from the same source. The motivation behind creating this model stemmed from two qualitative observations. First, the observable decline in the infection rate cannot be attributed to the exhaustion of the pool of susceptible healthy cells since the experimental evidence shows a relatively high percentage of cells remaining uninfected by the end of the experiment. Second, most cells, including infected ones, stay viable throughout the entire experiment.

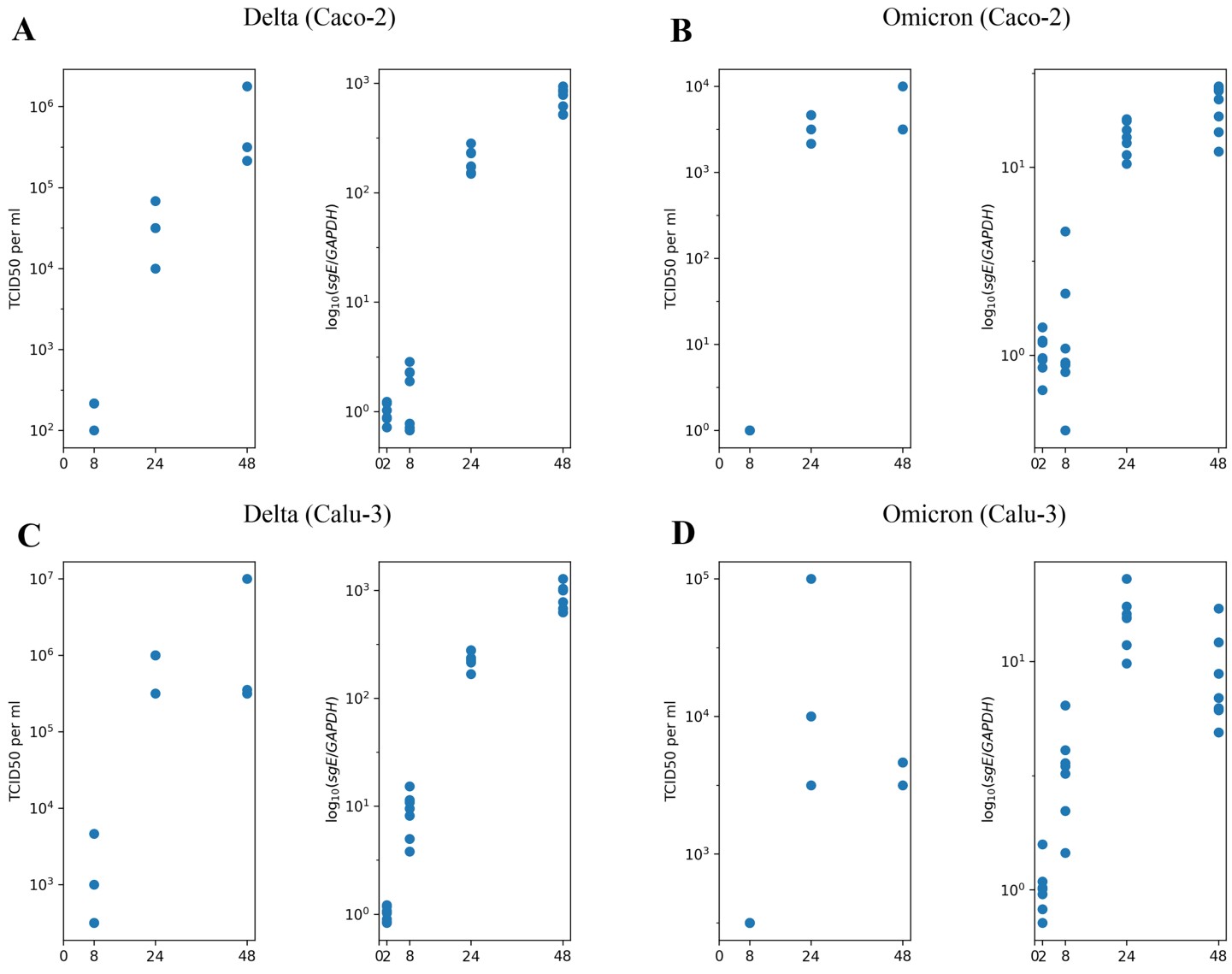

**Figure 1** **The experimental data employed in this study.** (A) Caco-2 cell line, the Delta variant of SARS-CoV-2; (B) Caco-2, the Omicron variant; (C) Calu-3, the Delta variant; (D) Calu-3, the Omicron variant. The left parts of panels (A–D) show $TCID_{50}$ assay values (infectious virus titers); the right parts show RT-qPCR results for subgenomic envelope viral RNA (*sgE*), normalized by the human housekeeping gene *GAPDH*.

We addressed these two issues by augmenting the classical infection model with an innate immune response and the consideration of cellular resource exhaustion, both essential components for understanding infection dynamics. A visual representation of our model is provided in Fig. 2. Briefly, a susceptible healthy cell becomes infected upon contact with a virus at a rate $\beta$. Following a latent period of $\Delta_{t\,latent}$ hours, an infected cell begins producing new viral particles as long as necessary cellular resources (*e.g.*, lipids) are not exhausted. Once these resources are depleted, virus production ceases, but the cell remains viable. An infected cell also releases cytokines at a constant rate $p_c$, starting $\Delta_{t\,cyt}$ hours post-infection. These cytokines subsequently interact with their receptors in healthy

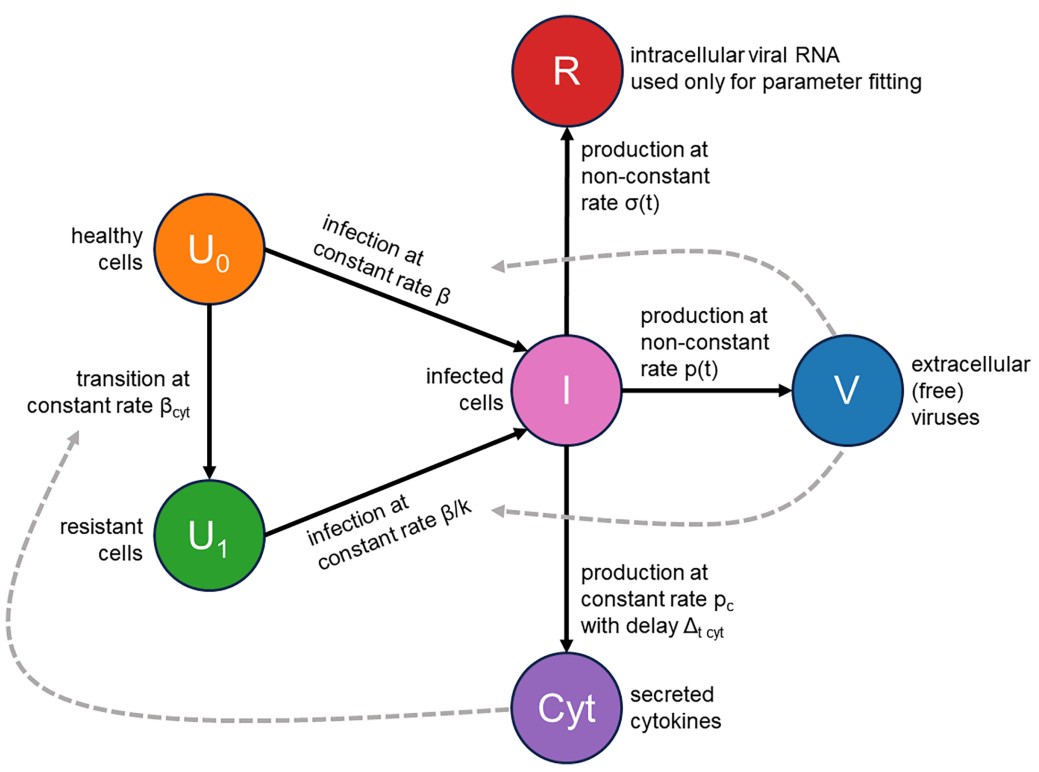

**Figure 2 Visual representation of the mathematical model.**

cells with an intensity of $\beta_{cyt}$ and make them immune by $k$-fold reduction of their viral entry rate. Additionally, we explicitly model the concentration of intracellular viral RNA, allowing us to directly use RT-qPCR data for parameter inference. The complete system of integro-differential equations is presented in the Materials and Methods section.

Herein, we employ the experimental measurements presented in Fig. 1 to estimate the model's parameter values for both the Delta and Omicron variants in Caco-2 and Calu-3 cells.

### Mathematical modeling suggests that the Omicron variant has a reduced viral entry rate and triggers a stronger immune response in Caco-2 cells compared to the Delta variant

Our parameter-fitting strategy was rooted in a dual objective. We aimed to align the model's output closely with the experimental data while also minimizing the number of variant-specific parameters to pinpoint the key factors underlying the observed differences. Current literature contains strong experimental evidence that infection rate and cytokine profiles drastically differ between the Omicron and previous variants (*Meng et al., 2022*; *Hu et al., 2022*; *Chan et al., 2022*; *Shuai et al., 2023*; *Singh et al., 2023*). Based on these data, we decided to link three parameters to the virus type: the infection rate $\beta$, the cytokine generation rate $p_c$, and the infectivity reduction rate $k$. Additionally, we made the

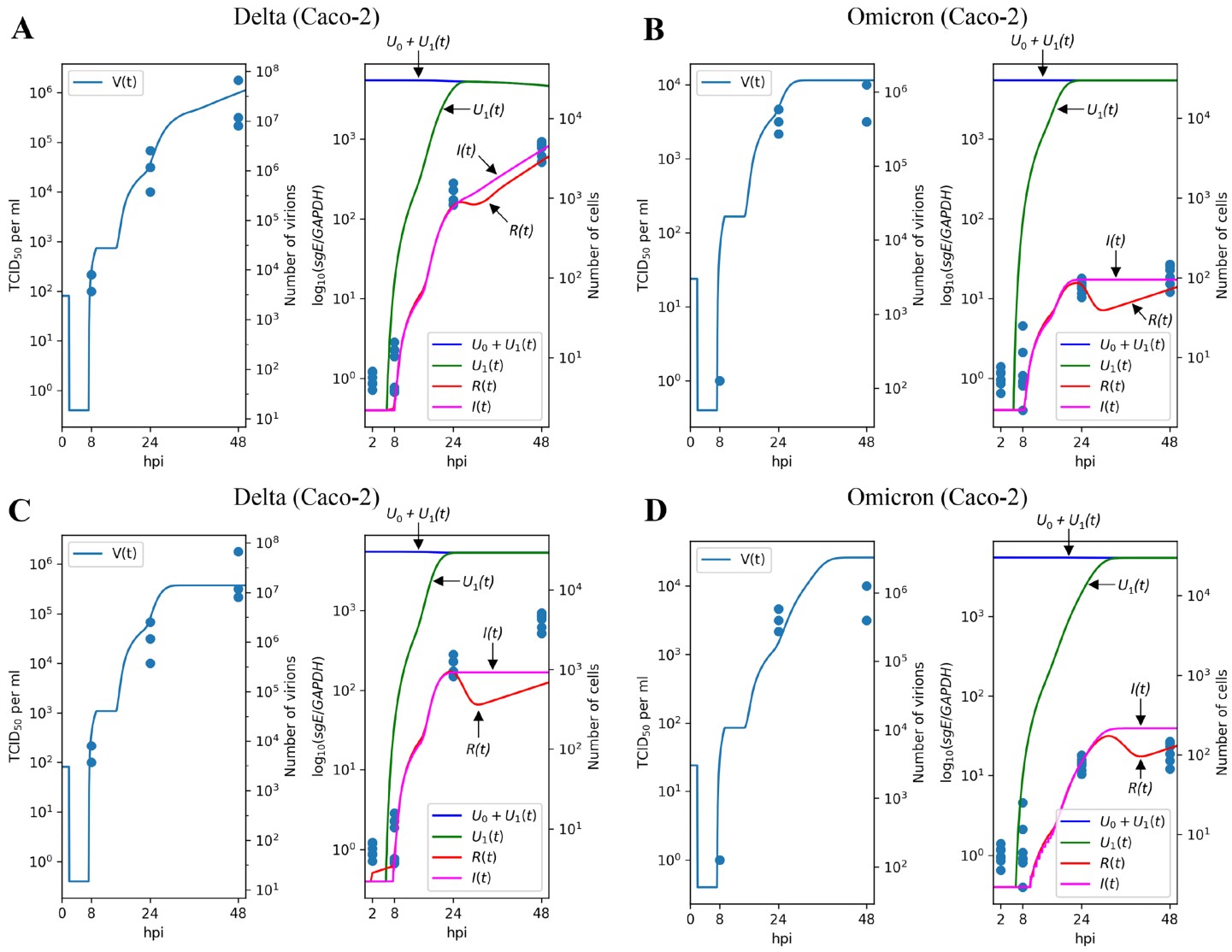

**Figure 3 Results of parameter fitting for Caco-2 cell line.** The top row shows the results for Delta (A) and Omicron (B) variants with three variant-specific parameters (the infection rate $\beta$, the cytokine generation rate $p_c$, and the infectivity reduction rate $k$). The bottom row shows the results for Delta (C) and Omicron (D) variants with a single variant-specific parameter (the infection rate $\beta$). The left parts of panels (A–D) show $TCID_{50}$ assay values (infectious virus titers); the right parts show RT-qPCR results for subgenomic envelope viral RNA ($sgE$), normalized by the human housekeeping gene *GAPDH*. The following parameter values were inferred for the bottom row: $\beta = 1.50455 \cdot 10^{-8}$ (Delta), $\beta = 3.97769 \cdot 10^{-9}$ (Omicron), $k = 5.27759 \cdot 10^{7}$, $p_c = 98375.9$.

translation of data from $TCID_{50}$ units into the number of virions ($Norm_V$ parameter) variant-specific, as Delta and Omicron variants exhibit varying infectivity levels.

The parameter fitting procedure is provided in the Materials and Methods section, and an exhaustive description of the model parameters and their values can be found in Table 1.

The results of parameter fitting for Caco-2 cells are depicted in Figs. 3A (Delta variant) and 3B (Omicron variant). It is evident that the model's output closely aligns with the experimental data, demonstrating a high level of precision. In qualitative terms, our model

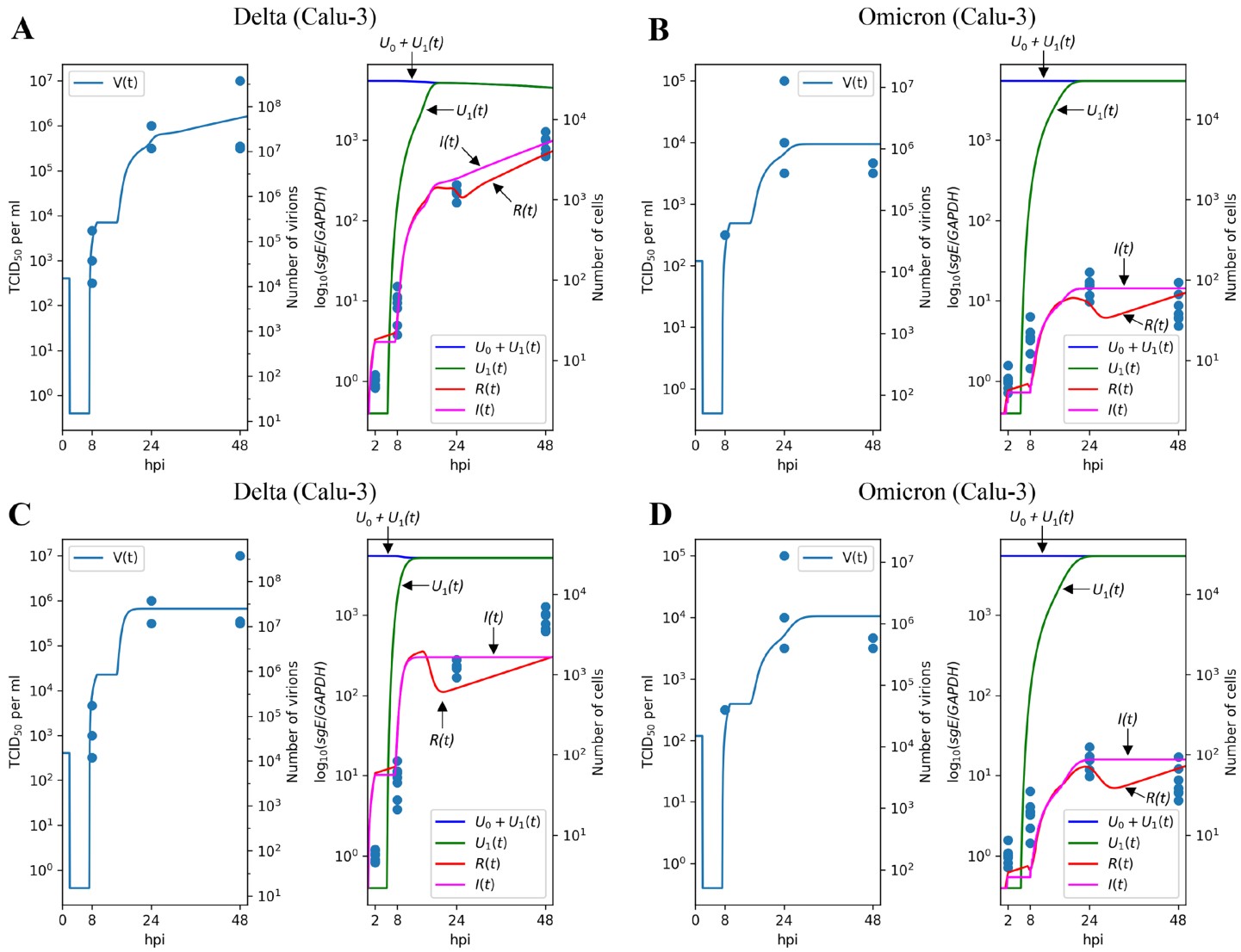

**Figure 4 Results of parameter fitting for Calu-3 cell line.** The top row shows the results for Delta (A) and Omicron (B) variants with three variant-specific parameters (the infection rate $\beta$, the cytokine generation rate $p_c$, and the infectivity reduction rate $k$). The bottom row shows the results for Delta (C) and Omicron (D) variants with a single variant-specific parameter (the infection rate $\beta$). The left parts of panels (A–D) show $TCID_{50}$ assay values (infectious virus titers); the right parts show RT-qPCR results for subgenomic envelope viral RNA ($sgE$), normalized by the human housekeeping gene $GAPDH$. The following parameter values were inferred for the bottom row: $\beta = 6.30644 \cdot 10^{-8}$ (Delta), $\beta = 3.68335 \cdot 10^{-9}$ (Omicron), $k = 3.29331 \cdot 10^7$, $p_c = 244798$.

reveals the following significant (in terms of confidence intervals) distinctions between Delta and Omicron variants:

- the viral entry rate $\beta$ in Caco-2 is approximately 1.3 times smaller for Omicron;

- the immune response is significantly stronger for Omicron, with the cytokine generation rate $p_c$ increasing by a factor of approximately 7.5 and the resistant cell protection $k$ growing by a factor of approximately $2.5 \cdot 10^7$. In fact, the value of $k$ for Omicron is equivalent to $+\infty$ which could be interpreted as the complete absence of infection of resistant cells.

Interestingly, attempting to account for the difference between Delta and Omicron variants by adjusting a single variant-specific parameter, the infection rate $\beta$, results in a significant reduction in accuracy: as seen in Figs. 3C and 3D, the model's curves deviate from a substantial number of experimental data points, and the model quality deteriorates by a factor of approximately 1.75, as assessed by the error functional. Thus, solely varying the infection rate is clearly insufficient to explain the distinction between Delta and Omicron.

Similar trends are observed when using only the cytokine generation rate $p_c$ as a variant-specific parameter, which leads to a quality reduction by a factor of approximately 1.1 (see Figs. S1A and S1B). Note that in the case of the Calu-3 cell line (see next section), the drop in fit quality is greater. When employing the infectivity reduction rate $k$ as the sole variant-specific parameter, the quality decreases by a factor of approximately 1.39 (see Figs. S1C and S1D).

## The results on the Delta and Omicron variants generalize to the Calu-3 cell line

To evaluate the robustness of the parameter fitting results, we extended our analysis to the Calu-3 cell line. Given that Caco-2 and Calu-3 represent distinct cell types with varying sets of expressed genes, we re-fitted specific cell-dependent parameters, including the infection rate $\beta$, the cytokine generation intensity $p_c$, and the reduction of infection intensity rate $k$, while preserving the values of the other ones.

Figures 4A and 4B illustrate the modeling results for the Delta and Omicron variants, respectively. Much like in the Caco-2 cell case, our model's predictions provide a reliable explanation for the experimental measurements. The differences between the Delta and Omicron parameters mirror those observed in Caco-2 cells: the entry rate $\beta$ is 4.3 times lower for the Omicron variant, the cytokine generation rate $p_c$ is 5.4 times higher, and the protection rate of resistant cells, $k$, is increased by a factor of $7 \cdot 10^6$ (as in the previous case, this is almost equivalent to $k = +\infty$ for Omicron).

Concordant with the Caco-2 case, variation of only one parameter essentially reduces approximation quality: by factor 1.38 for $\beta$ (Figs. 4C and 4D), 1.15 for $p_c$ (Figs. S2A and S2B), and 1.81 for $k$ (Figs. S2C and S2D).

## DISCUSSION

Mathematical modeling is a potent tool for understanding complex time-dependent biological processes (*MacArthur, Stumpf & Oreffo, 2020*). By merging experimental data with a theoretical framework, these models provide quantitative insights into viral infection dynamics (*Desikan et al., 2022*). Such modeling is invaluable for studying SARS-CoV-2 within-host dynamics (*Prague et al., 2022*; *Hernandez-Vargas & Velasco-Hernandez, 2020*; *Rodriguez & Dobrovolny, 2021*). Specifically, mathematical models of viral propagation within cells, tissues, and organs assess intricate processes that are difficult or impossible to directly monitor and measure through experimental means (*Graw & Perelson, 2016*). While many studies focus on modeling within-host dynamics at the level of the whole organism (*Jeong et al., 2021*; *Ejima et al., 2021*; *Iwanami et al., 2021*), it is

equally important to model the viral cycle at the level of individual susceptible cell types (*Staroverov et al., 2023*; *Bernhauerová et al., 2021*).

The results obtained by applying our mathematical model highlight the significance of three key parameters in fitting the experimental data of SARS-CoV-2 variant dynamics in cells. While some studies propose that the infectivity rate is the primary factor distinguishing the Omicron and Delta variants, our findings demonstrate that one should consider three parameters, including those related to the immune response, to achieve a reliable fit between the model and the experimental data. Among all Omicron proteins, S-protein and nsp6 play the most significant role in viral attenuation compared to previous SARS-CoV-2 variants (*Chen et al., 2023*). While S-protein mutations define the infectivity rate, nsp6 activates NLR3-dependent cytokine production and pyroptosis (*Sun et al., 2022*), suggesting an essential role of innate immunity in Omicron attenuation.

Among three key parameters of our model, the infection rate $\beta$ was consistently lower for the Omicron variant compared to the Delta variant in both Caco-2 and Calu-3 cells. The SARS-CoV-2 S protein binds the ACE2 receptor, triggering cleavage by cellular proteases for viral cell entry. Omicron variant's S protein has higher ACE2 affinity than Delta, but its cleavage efficiency is reduced (*Meng et al., 2022*). Moreover, Omicron S protein mutations have hindered the effective utilization of TMPRSS2 protease compared to Delta. Consequently, this change has shifted the main cellular entry route from membrane fusion to endocytosis (*Meng et al., 2022*). In an experiment using TMPRSS2-overexpressing VeroE6 cells, the TMPRSS2 inhibitor camostat markedly hindered Delta variant cell entry, but minimally affected Omicron entry. In contrast, chloroquine and bafilomycin A1, inhibitors of the endocytic pathway, reduced cell entry for both variants (*Zhao et al., 2022*). Omicron demonstrated about tenfold higher infectivity in TMPRSS2-lacking HEK cells than Delta. Omicron exhibited sensitivity to E64d, an inhibitor of cathepsins B and L, while being insensitive to camostat. These findings collectively suggest Omicron's preference for the endocytic pathway (*Willett et al., 2022*). Omicron showed lower infectivity in Calu-3 and Caco-2 cells compared to Delta; notably, the proportion of infected cells differed about two fold or more at 48 and 72 h post-infection in the Caco-2 cell line (*Bojkova et al., 2022b*).

The second variant-specific factor in our model for the SARS-CoV-2 in Caco-2 and Calu-3 cells is the cytokine generation intensity $p_c$. The model suggests that this parameter should be higher for the Omicron variant in both cell lines. Endosomal Toll-like receptors (TLR3/7/8) respond to SARS-CoV-2 presence by initiating signaling, augmenting secretion of type I and III interferons (IFNs) and proinflammatory cytokines (*Zhou et al., 2022*). Presumably, Omicron's delayed entry and endocytic pathway use elevate extracellular IFN secretion. Interaction of type I and III IFN receptors with their ligands leads to phosphorylation of STAT1 and STAT2, forming a complex with IRF9, known as IFN-stimulated gene factor 3 (ISGF3). ISGF3 acts as a transcriptional activator for IFN-stimulated genes. While type I and III IFNs share signaling pathways, type I IFNs are expressed earlier and bind receptors on most cells, while type III IFN receptors are found mainly on epithelial cells and select myeloid lineage leukocytes. Consequently, the response to type I IFNs is often more rapid and robust, leading to the production of

additional proinflammatory cytokines and chemokines. Uncontrolled stimulation, however, risks excessive host cell damage (*Schoggins, 2019*).

The higher cytokine generation intensity suggested for the Omicron variant compared to the Delta variant in our model is substantiated by the data from other studies. SARS-CoV-2 Omicron isolates, unlike Delta, triggered IFN pathways, evidenced by IRF promoter activation in A549 cells (*Bojkova et al., 2022b*). Omicron also raised STAT1 phosphorylation more than Delta did in Caco-2 and Calu-3 cells, a crucial IFN response event (*Bojkova et al., 2022a*). In Calu-3 cells, Omicron infection prompted notably higher IFN-β and epithelial-specific IFN-λ1 mRNA induction, along with their target genes *ISG15* and *OAS1*, as early as 12 h post-infection compared to the Delta variant (B.1.617.2) (*Singh et al., 2023*). One study reported elevated cytokine expression in Calu-3 cells infected with Omicron variants BA.1 and BA.2 relative to Delta: *IFNB* and *CXCL10* gene expression was about four-fold higher, and secreted IFN-β and CXCL10 levels were approximately three-fold higher at 48 h post-infection (*Reuschl et al., 2024*). Another study observed significantly higher IFN-β secretion from Calu-3 cells infected with Omicron variant BA.2 compared to the Wuhan variant 72 h post-infection (*Gori Savellini, Anichini & Cusi, 2023*).

The third variant-specific parameter in our model is the infectivity reduction rate $k$, representing the enhanced virus resistance of cells primed by cytokines. To align the model with the experimental data, this parameter must be higher for the Omicron variant compared to the Delta variant. Viral entry *via* the endosomal pathway renders SARS-CoV-2 susceptible to IFN-inducible inhibitory molecules, such as IFITM2/3 (*Peacock et al., 2021*; *Winstone et al., 2021*) or NCOA7 (*Khan et al., 2021*). In this context, the predominant utilization of the endocytic pathway by the Omicron variant may elucidate why the entry kinetics of Omicron into cells that are already "primed" by IFNs would be slower than that of the Delta variant. LY6E, an IFN-inducible membrane-anchored protein, has been shown to impede SARS-CoV-2 cell entry, while IFIT family proteins have been implicated in the inhibition of SARS-CoV-2 replication (*Martin-Sancho et al., 2021*; *Pfaender et al., 2020*), further corroborating the theory of viral inhibition during preferential employment of the endocytic route. In Vero cells, lacking a functional IFN response, no discernible disparities in the proportion of infected cells were observed between Omicron and Delta variant infections. Conversely, in immune-competent Caco-2 and Calu-3 cells, the Omicron variant infected fewer cells than the Delta variant (*Bojkova et al., 2022b*). Additionally, in A549 cells engineered to express ACE2 and TMPRSS2, Omicron exhibited reduced infection capacity compared to Delta. However, these distinctions were nullified upon suppression of pattern recognition receptors MDA5 and RIG-I, both of which contribute to the cellular IFN response, suggesting enhanced cellular resilience to Omicron following IFN activation (*Bojkova et al., 2022a*). Furthermore, in Caco-2 cells, the Omicron variant displayed considerably heightened sensitivity to treatment with IFN-α, -β, and -γ compared to the Delta variant (*Bojkova et al., 2022a*).

Based on the above, our model suggests that the differences in the infection dynamics of intestinal and lung cells by Omicron and Delta variants are influenced not only by varying

rates of virus entry but also by differences in cytokine secretion intensity and increased cell resistance to the virus following cytokine activation.

Our study has several limitations. Firstly, while Omicron shows attenuated replication in the intestinal and lung cells, it replicates more efficiently than Delta and wild-type SARS-CoV-2 in human nasal epithelial cells (*Shuai et al., 2023*). The differences in replication fitness of Omicron in the upper and lower respiratory tracts are yet to be revealed, and this comprises an important direction of our future research. Secondly, we did not use any direct measurements of immune response-related variables (*e.g.*, cytokine expression over time), so the corresponding parameter values refer to abstract relative units and should be interpreted only in the context of between-variant comparisons. Lastly, the data used for parameter fitting contain few time points and thus large time intervals have no experimental observations. As a result, the model predictions indeed have a significant degree of uncertainty in these time intervals.

## ACKNOWLEDGEMENTS

The authors thank Dr. Stepan Nersisyan and Dr. Hin Chu for their valuable comments and discussions.

### Funding

The research was performed within the framework of the Basic Research Program at HSE University. The funders had no role in study design, data collection and analysis, decision to publish, or preparation of the manuscript.

### Grant Disclosures

The following grant information was disclosed by the authors:
Basic Research Program at HSE University.

### Competing Interests

Alexander Tonevitsky is employed by Art Photonics GmbH.

### Author Contributions

- Vladimir Staroverov conceived and designed the experiments, performed the experiments, analyzed the data, prepared figures and/or tables, authored or reviewed drafts of the article, and approved the final draft.
- Alexei Galatenko conceived and designed the experiments, performed the experiments, analyzed the data, prepared figures and/or tables, authored or reviewed drafts of the article, and approved the final draft.
- Evgeny Knyazev conceived and designed the experiments, performed the experiments, analyzed the data, prepared figures and/or tables, authored or reviewed drafts of the article, and approved the final draft.
- Alexander Tonevitsky conceived and designed the experiments, authored or reviewed drafts of the article, and approved the final draft.

## Data Availability

The full model description and parameter values are available in the main text and Supplemental File.

## Supplemental Information

Supplemental information for this article can be found online at http://dx.doi.org/10.7717/peerj.16964#supplemental-information.

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
