# Peer review of "Mathematical model explains differences in Omicron and Delta SARS-CoV-2 dynamics in Caco-2 and Calu-3 cells"

_PeerJ, doi:10.7717/peerj.16964_

## Round 0.1 · original submission · Major Revisions

Please address the concerns of both reviewers and mend the manuscript accordingly. To address critiques of Reviewer #2 (who recommended rejection), some discussion should be added on the general importance of mathematical modeling of biological and pathological processes.

Reviewer 1 ·

Basic reporting

The manuscript meets the basic reporting requirements well. I particularly appreciate the care the authors took in mapping their modeling results to supporting experimental work in the discussion.

Experimental design

There are some issues with the methodology and description of the methodology:
1. The model as written has a unit issue --- using the same beta value in equations 1 and 5 means that virus and cells are measured in the same units or tacitly assumes that one virion enters a cell to cause infection. There is a similar issue with beta_cyt --- since it has the same value in equations 1 and 3, cells and cytokines are expressed in the same unit. Is this what the authors intended?
2. What is pcs (used as a unit in Table 1)?
3. Some of the parameters in Table 1 do not seem to be explained in the text --- (Norm parameters, R0, alpha, alpha')
4. How was it decided that beta, p, and k would vary between strains? Were other parameters or parameter combinations tested?
5. I think it would be helpful to include the error functionals.
6. Why are there several possible values alpha and alpha'? How were there used in fitting? What are the values used for the fits shown in the manuscript?
7. I think it would be helpful to include the model diagram in this manuscript.

Validity of the findings

As noted above, I am concerned about the structure of the model --- particularly the assumption that beta has the same value in equations 1 and 5. Correcting this might change the results. Otherwise, I believe the fitting is done correctly and based on those results, the conclusions drawn are reasonable.

Reviewer 2 ·

Basic reporting

The authors established a mathematic model by validating RNA-seq data that illustrated the infectiveness and immune response triggered by delta and omicron SARS-CoV-2 in cell lines. It is confirmed what we observed in real world, however add no value to the understanding of SARS-CoV-2 biology and COVID-19 treatment. Authors did not give any potential target by this model. Simplely by RNA-Seq itself, researchers and doctors can draw the same conclusion without mathemetic model. So I would suggest authors deeply investigate their data, and try to establish a model that influence clinical desicion for the treatment.

Experimental design

no comment

Validity of the findings

no comment

---

## Round 0.2 · accepted · Accept

All concerns of the reviewers were addressed and the manuscript was amended accordingly.

Reviewer 1 ·

Basic reporting

No comment

Experimental design

No comment

Validity of the findings

No comment

Additional comments

The authors have addressed all my previous comments to my satisfaction.